# A Likelihood-Free Inference Framework for Population Genetic Data using Exchangeable Neural Networks

**Jeffrey Chan**
University of California, Berkeley
chanjed@berkeley.edu

**Valerio Perrone**
University of Warwick
v.perrone@warwick.ac.uk

**Jeffrey P. Spence**
University of California, Berkeley
spence.jeffrey@berkeley.edu

**Paul A. Jenkins**
University of Warwick
p.jenkins@warwick.ac.uk

**Sara Mathieson**
Swarthmore College
smathie1@swarthmore.edu

**Yun S. Song**
University of California, Berkeley
yss@berkeley.edu

## Abstract

An explosion of high-throughput DNA sequencing in the past decade has led to a surge of interest in population-scale inference with whole-genome data. Recent work in population genetics has centered on designing inference methods for relatively simple model classes, and few scalable general-purpose inference techniques exist for more realistic, complex models. To achieve this, two inferential challenges need to be addressed: (1) population data are exchangeable, calling for methods that efficiently exploit the symmetries of the data, and (2) computing likelihoods is intractable as it requires integrating over a set of correlated, extremely high-dimensional latent variables. These challenges are traditionally tackled by likelihood-free methods that use scientific simulators to generate datasets and reduce them to hand-designed, permutation-invariant summary statistics, often leading to inaccurate inference. In this work, we develop an exchangeable neural network that performs summary statistic-free, likelihood-free inference. Our framework can be applied in a black-box fashion across a variety of simulation-based tasks, both within and outside biology. We demonstrate the power of our approach on the recombination hotspot testing problem, outperforming the state-of-the-art.

## 1 Introduction

Statistical inference in population genetics aims to quantify the evolutionary events and parameters that led to the genetic diversity we observe today. Population genetic models are typically based on the coalescent [1], a stochastic process describing the distribution over genealogies of a random exchangeable set of DNA sequences from a large population. Inference in such complex models is challenging. First, standard coalescent-based likelihoods require integrating over a large set of correlated, high-dimensional combinatorial objects, rendering classical inference techniques inapplicable. Instead, likelihoods are implicitly defined via scientific simulators (i.e., generative models), which draw a sample of correlated trees and then model mutation as Poisson point processes on the sampled trees to generate sequences at the leaves. Second, inference demands careful treatment

of the exchangeable structure of the data (a set of sequences), as disregarding it leads to an exponential increase in the already high-dimensional state space.

Current likelihood-free methods in population genetics leverage scientific simulators to perform inference, handling the exchangeable-structured data by reducing it to a suite of low-dimensional, permutation-invariant summary statistics [2, 3]. However, these hand-engineered statistics typically are not statistically sufficient for the parameter of interest. Instead, they are often based on the intuition of the user, need to be modified for each new task, and are not amenable to hyperparameter optimization strategies since the quality of the approximation is unknown.

The goal of this work is to develop a general-purpose inference framework for raw population genetic data that is not only likelihood-free, but also summary statistic-free. We achieve this by designing a neural network that exploits data exchangeability to learn functions that accurately approximate the posterior. While deep learning offers the possibility to work directly with genomic sequence data, poorly calibrated posteriors have limited its adoption in scientific disciplines [4]. We overcome this challenge with a training paradigm that leverages scientific simulators and repeatedly draws fresh samples at each training step. We show that this yields calibrated posteriors and argue that, under a likelihood-free inference setting, deep learning coupled with this 'simulation-on-the-fly' training has many advantages over the more commonly used Approximate Bayesian Computation (ABC) [2, 5]. To our knowledge, this is the first method that handles the raw exchangeable data in a likelihood-free context.

As a concrete example, we focus on the problems of recombination hotspot testing and estimation. Recombination is a biological process of fundamental importance, in which the reciprocal exchange of DNA during cell division creates new combinations of genetic variants. Experiments have shown that many species exhibit *recombination hotspots*, i.e., short segments of the genome with high recombination rates [6]. The task of recombination hotspot testing is to predict the location of recombination hotspots given genetic polymorphism data. Accurately localizing recombination hotspots would illuminate the biological mechanism that underlies recombination, and could help geneticists map the mutations causing genetic diseases [7]. We demonstrate through experiments that our proposed framework outperforms the state-of-the-art on the hotspot detection problem.

Our main contributions are:

- A novel exchangeable neural network that respects permutation invariance and maps from the data to the posterior distribution over the parameter of interest.

- A simulation-on-the-fly training paradigm, which leverages scientific simulators to achieve calibrated posteriors.

- A general-purpose likelihood-free Bayesian inference method that combines the exchangeable neural network and simulation-on-the-fly training paradigm to both discrete and continuous settings. Our method can be applied to many population genetic settings by making straightforward modifications to the simulator and the prior, including demographic model selection, archaic admixture detection, and classifying modes of natural selection.

- An application to a single-population model for recombination hotspot testing and estimation, outperforming the model-based state-of-the-art, `LDhot`. Our approach can be seamlessly extended to more complex model classes, unlike `LDhot` and other model-based methods.

Our software package `defiNETti` is publicly available at `https://github.com/popgenmethods/defiNETti`.

## 2    Related Work

Likelihood-free methods like ABC have been widely used in population genetics [2, 5, 8, 9, 10]. In ABC the parameter of interest is simulated from its prior distribution, and data are subsequently simulated from the generative model and reduced to a pre-chosen set of summary statistics. These statistics are compared to the summary statistics of the real data, and the simulated parameter is weighted according to the similarity of the statistics to derive an empirical estimate of the posterior distribution. However, choosing summary statistics for ABC is challenging because there is a trade-off between loss of sufficiency and computational tractability. In addition, there is no direct way to evaluate the accuracy of the approximation.

Other likelihood-free approaches have emerged from the machine learning community and have been applied to population genetics, such as support vector machines (SVMs) [11, 12], single-layer neural networks [13], and deep learning [3]. Recently, a (non-exchangeable) convolutional neural network method was proposed for raw population genetic data [14]. The connection between likelihood-free Bayesian inference and neural networks has also been studied previously [15, 16]. An attractive property of these methods is that, unlike ABC, they can be applied to multiple datasets without repeating the training process (i.e., amortized inference). However, current practice in population genetics collapses the data to a set of summary statistics before passing it through the machine learning models. Therefore, the performance still rests on the ability to laboriously hand-engineer informative statistics, and must be repeated from scratch for each new problem setting.

The inferential accuracy and scalability of these methods can be improved by exploiting symmetries in the input data. Permutation-invariant models have been previously studied in machine learning for SVMs [17] and recently gained a surge of interest in the deep learning literature. Recent work on designing architectures for exchangeable data include [18], [19], and [20], which exploit parameter sharing to encode invariances.

We demonstrate these ideas on the discrete and continuous problems of recombination hotspot testing and estimation, respectively. To this end, several methods have been developed (see, e.g., [21, 22, 23] for the hotspot testing problem). However, none of these are scalable to the whole genome, with the exception of `LDhot` [24, 25], so we limit our comparison to this latter method. `LDhot` relies on a composite likelihood, which can be seen as an approximate likelihood for summaries of the data. It can be computed only for a restricted set of models (i.e., an unstructured population with piecewise constant population size), is unable to capture dependencies beyond those summaries, and scales at least cubically with the number of DNA sequences. The method we propose in this paper scales linearly in the number of sequences while using raw genetic data directly.

# 3 Methods

## 3.1 Problem Setup

Likelihood-free methods use coalescent simulators to draw parameters from the prior $\theta^{(i)} \sim \pi(\theta)$ and then simulate data according to the coalescent $\mathbf{x}^{(i)} \sim \mathbb{P}(\mathbf{x} \mid \theta^{(i)})$, where $i$ is the index of each simulated dataset. Each population genetic datapoint $\mathbf{x}^{(i)} \in \{0, 1\}^{n \times d}$ typically takes the form of a binary matrix, where rows correspond to individuals and columns indicate the presence of a Single Nucleotide Polymorphism (SNP), a variable site in a DNA sequence[1]. Our goal is to learn the posterior $\mathbb{P}(\theta \mid \mathbf{x}_{obs})$, where $\theta$ is the parameter of interest and $\mathbf{x}_{obs}$ is the observed data. For unstructured populations the order of individuals carries no information, hence the rows are exchangeable. More concretely, given data $\mathbf{X} = (\mathbf{x}^{(1)}, \ldots \mathbf{x}^{(N)})$ where $\mathbf{x}^{(i)} := (x_1^{(i)}, \ldots, x_n^{(i)}) \sim \mathbb{P}(\mathbf{x} \mid \theta^{(i)})$ and $x_j^{(i)} \in \{0, 1\}^d$, we call $\mathbf{X}$ *exchangeably-structured* if, for every $i$, the distribution over the rows of a single datapoint is permutation-invariant

$$\mathbb{P}\left(x_1^{(i)}, \ldots, x_n^{(i)} \mid \theta^{(i)}\right) = \mathbb{P}\left(x_{\sigma(1)}^{(i)}, \ldots, x_{\sigma(n)}^{(i)} \mid \theta^{(i)}\right),$$

for all permutations $\sigma$ of the indices $\{1, \ldots, n\}$. For inference, we propose iterating the following algorithm.

1. *Simulation-on-the-fly*: Sample a fresh minibatch of $\theta^{(i)}$ and $\mathbf{x}^{(i)}$ from the prior and coalescent simulator.

2. *Exchangeable neural network*: Learn the posterior $\mathbb{P}(\theta^{(i)} \mid \mathbf{x}^{(i)})$ via an exchangeable mapping with $\mathbf{x}^{(i)}$ as the input and $\theta^{(i)}$ as the label.

This framework can then be applied to learn the posterior of the evolutionary model parameters given $\mathbf{x}_{obs}$. The details on the two building blocks of our method, namely the exchangeable neural network and the simulation-on-the-fly paradigm, are given in Section 3.2 and 3.3, respectively.

## 3.2 Exchangeable Neural Network

The goal of the exchangeable neural network is to learn the function $f : \{0,1\}^{n \times d} \to \mathcal{P}_\Theta$, where $\Theta$ is the space of all parameters $\theta$ and $\mathcal{P}_\Theta$ is the space of all probability distributions on $\Theta$. We parameterize the exchangeable neural network by applying the same function to each row of the binary matrix, then applying a symmetric function to the output of each row, finally followed by yet another function mapping from the output of the symmetric function to a posterior distribution. More concretely,

$$f(\mathbf{x}) := (h \circ g)\big(\Phi(x_1), \ldots, \Phi(x_n)\big),$$

where $\Phi : \{0,1\}^d \to \mathbb{R}^{d_1}$ is a function parameterized by a convolutional neural network, $g : \mathbb{R}^{n \times d_1} \to \mathbb{R}^{d_2}$ is a symmetric function, and $h : \mathbb{R}^{d_2} \to \mathcal{P}_\Theta$ is a function parameterized by a fully connected neural network. A variant of this representation is proposed by [18] and [20]. See Figure 1 for an example. Throughout the paper, we choose $g$ to be the mean of the element-wise top decile, such that $d_1 = d_2$ in order to allow for our method to be robust to changes in $n$ at test time. Many other symmetric functions such as the element-wise sum, element-wise max, lexicographical sort, or higher-order moments can be employed.

This exchangeable neural network has many advantages. While it could be argued that flexible machine learning models could learn the structured exchangeability of the data, encoding exchangeability explicitly allows for faster per-iteration computation and improved learning efficiency, since data augmentation for exchangeability scales as $O(n!)$. Enforcing exchangeability implicitly reduces the size of the input space from $\{0,1\}^{n \times d}$ to the quotient space $\{0,1\}^{n \times d}/S_n$, where $S_n$ is the symmetric group on $n$ elements. A factorial reduction in input size leads to much more tractable inference for large $n$. In addition, choices of $g$ where $d_2$ is independent of $n$ (e.g., quantile operations with output dimension independent of $n$) allows for an inference procedure which is robust to differing number of exchangeable variables between train and test time. This property is particularly desirable for performing inference with missing data.

## 3.3 Simulation-on-the-fly

Supervised learning methods traditionally use a fixed training set and make multiple passes over the data until convergence. This training paradigm typically can lead to a few issues: poorly calibrated posteriors and overfitting. While the latter has largely been tackled by regularization methods and large datasets, the former has not been sufficiently addressed. We say a posterior is calibrated if for $X_{q,A} := \{\mathbf{x} \mid \hat{p}(\theta \in A \mid \mathbf{x}) = q\}$, we have $\mathbb{E}_{\mathbf{x} \in X_{q,A}}[p(\theta \in A \mid \mathbf{x})] = q$ for all $q, A$. Poorly calibrated posteriors are particularly an issue in scientific disciplines as scientists often demand methods with calibrated uncertainty estimates in order to measure the confidence behind new scientific discoveries (often leading to reliance on traditional methods with asymptotic guarantees such as MCMC).

When we have access to scientific simulators, the amount of training data available is limited only by the amount of compute time available for simulation, so we propose simulating each training datapoint afresh such that there is exactly one epoch over the training data (i.e., no training point is passed through the neural network more than once). We refer to this as *simulation-on-the-fly*. Note that this can be relaxed to pass each training point a small constant number of times in the case of computational constraints on the simulator. This approach guarantees properly calibrated posteriors and obviates the need for regularization techniques to address overfitting. Below we justify these properties through the lens of statistical decision theory.

More formally, define the Bayes risk for prior $\pi(\theta)$ as $R_\pi^* = \inf_T \mathbb{E}_{\mathbf{x}} \mathbb{E}_{\theta \sim \pi}[l(\theta, T(\mathbf{x})]$, with $l$ being the loss function and $T$ an estimator. The excess risk over the Bayes risk resulting from an algorithm $A$ with model class $\mathcal{F}$ can be decomposed as

$$R_\pi(\tilde{f}_A) - R_\pi^* = \underbrace{\Big(R_\pi(\tilde{f}_A) - R_\pi(\hat{f})\Big)}_{\text{optimization error}} + \underbrace{\Big(R_\pi(\hat{f}) - \inf_{f \in \mathcal{F}} R_\pi(f)\Big)}_{\text{estimation error}} + \underbrace{\Big(\inf_{f \in \mathcal{F}} R_\pi(f) - R_\pi^*\Big)}_{\text{approximation error}},$$

where $\tilde{f}_A$ and $\hat{f}$ are the function obtained via algorithm $A$ and the empirical risk minimizer, respectively. The terms on the right hand side are referred to as the optimization, estimation, and approximation errors, respectively. Often the goal of statistical decision theory is to minimize the excess risk motivating algorithmic choices to control the three sources of error. For example, with

supervised learning, overfitting is a result of large estimation error. Typically, for a sufficiently expressive neural network optimized via stochastic optimization techniques, the excess risk is dominated by optimization and estimation errors. Simulation-on-the-fly guarantees that the estimation error is small, and as neural networks typically have small approximation error, we can conclude that the main source of error remaining is the optimization error. It has been shown that smooth population risk surfaces can induce jagged empirical risk surfaces with many local minima [26, 27]. We confirmed this phenomenon empirically in the population genetic setting(Section 5) showing that the risk surface is much smoother in the on-the-fly setting than the fixed training setting. This reduces the number of poor local minima and, consequently, the optimization error. The estimator corresponding to the Bayes risk (for the cross-entropy or KL-divergence loss function) is the posterior. Thus, the simulation-on-the-fly training paradigm guarantees generalization and calibrated posteriors (assuming small optimization error).

# 4  Statistical Properties

The most widely-used likelihood-free inference method is ABC. In this section we briefly review ABC and show that our method exhibits the same theoretical guarantees together with a set of additional desirable properties.

**Properties of ABC**   Let $\mathbf{x}_{obs}$ be the observed dataset, $S$ be the summary statistic, and $d$ be a distance metric. The algorithm for vanilla rejection ABC is as follows. Denoting by $i$ each simulated dataset, for $i = 1 \ldots N$,

1. Simulate $\theta^{(i)} \sim \pi(\theta)$ and $\mathbf{x}^{(i)} \sim \mathbb{P}(\mathbf{x} \mid \theta^{(i)})$
2. Keep $\theta^{(i)}$ if $d(S(\mathbf{x}^{(i)}), S(\mathbf{x}_{obs})) \leq \epsilon$.

The output provides an empirical estimate of the posterior. Two key results regarding ABC make it an attractive method for Bayesian inference: (1) **Asymptotic guarantee:** As $\epsilon \to 0$, $N \to \infty$, and if $S$ is sufficient, the estimated posterior converges to the true posterior (2) **Calibration of ABC:** A variant of ABC (noisy ABC in [28]) which injects noise into the summary statistic function is calibrated. For detailed proofs as well as more sophisticated variants, see [28]. Note that ABC is notoriously difficult to perform diagnostics on without the ground truth posterior as many factors could contribute to a poor posterior approximation: poor choice of summary statistics, incorrect distance metric, insufficient number of samples, or large $\epsilon$.

**Properties of Our Method**   Our method matches both theoretical guarantees of ABC — (1) asymptotics and (2) calibration — while also exhibiting additional properties: (3) amortized inference, (4) no dependence on user-defined summary statistics, and (5) straightforward diagnostics. While the independence of summary statistics and calibration are theoretically justified in Section 3.2 and 3.3, we provide some results that justify the asymptotics, amortized inference, and diagnostics.

In the simulation-on-the-fly setting, convergence to a global minimum implies that a sufficiently large neural network architecture represents the true posterior within $\epsilon$-error in the following sense: for any fixed error $\epsilon$, there exist $H_0$ and $N_0$ such that the trained neural network produces a posterior which satisfies

$$\min_{\mathbf{w}} \mathbb{E}_{\mathbf{x}}\Big[ KL\Big( \mathbb{P}(\theta \mid \mathbf{x}) \,\big\|\, \mathbb{P}_{DL}^{(N)}(\theta \mid \mathbf{x}; \mathbf{w}, H) \Big) \Big] < \epsilon, \tag{1}$$

for all $H > H_0$ and $N > N_0$, where $H$ is the minimum number of hidden units across all neural network layers, $N$ is the number of training points, $\mathbf{w}$ the weights parameterizing the network, and KL the Kullback–Leibler divergence between the population risk and the risk of the neural network. Under these assumptions, the following proposition holds.

**Proposition 1.** *For any* $\mathbf{x}$, $\epsilon > 0$, *and fixed error* $\delta > 0$, *there exists an* $H > H_0$, *and* $N > N_0$ *such that,*

$$KL\Big( \mathbb{P}(\theta \mid \mathbf{x}) \,\big\|\, \mathbb{P}_{DL}^{(N)}(\theta \mid \mathbf{x}; \mathbf{w}^*, H) \Big) < \delta \tag{2}$$

*with probability at least* $1 - \frac{\epsilon}{\delta}$, *where* $\mathbf{w}^*$ *is the minimizer of* (1).

We can get stronger guarantees in the discrete setting common to population genetic data.

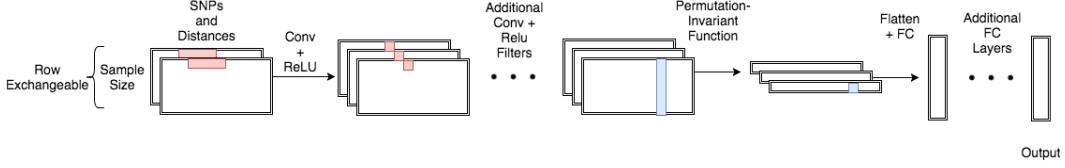

Figure 1: A cartoon schematic of the exchangeable architecture for population genetics.

**Corollary 1.** *Under the same conditions, if* $\mathbf{x}$ *is discrete and* $\mathbb{P}(\mathbf{x}) > 0$ *for all* $\mathbf{x}$*, the KL divergence appearing in* (2) *converges to* $0$ *uniformly in* $\mathbf{x}$*, as* $H, N \to \infty$.

The proofs are given in the supplement. These results exhibit both the asymptotic guarantees of our method and show that such guarantees hold for all $\mathbf{x}$ (i.e. amortized inference). Diagnostics for the quality of the approximation can be performed via hyperparameter optimization to compare the relative loss of the neural network under a variety of optimization and architecture settings.

## 5 Empirical Study: Recombination Hotspot Testing

In this section, we study the accuracy of our framework to test for recombination hotspots. As very few hotspots have been experimentally validated, we primarily evaluate our method on simulated data, with parameters set to match human data. The presence of ground truth allows us to benchmark our method and compare against `LDhot` (additional details on `LDhot` in the supplement). For the posterior in this classification task (hotspot or not), we use the softmax probabilities. Unless otherwise specified, for all experiments we use the mutation rate, $\mu = 1.1 \times 10^{-8}$ per generation per nucleotide, convolution patch length of 5 SNPs, 32 and 64 convolution filters for the first two convolution layers, 128 hidden units for both fully connected layers, and 20-SNP length windows. The experiments comparing against `LDhot` used sample size $n = 64$ to construct lookup tables for `LDhot` quickly. All other experiments use $n = 198$, matching the size of the CEU population (i.e., Utah Residents with Northern and Western European ancestry) in the 1000 Genomes dataset. All simulations were performed using `msprime` [29]. Gradient updates were performed using Adam [30] with learning rate $1 \times 10^{-3} \times 0.9^{b/10000}$, $b$ being the batch count. In addition, we augment the binary matrix, $\mathbf{x}$, to include the distance information between neighboring SNPs in an additional channel resulting in a tensor of size $n \times d \times 2$.

### 5.1 Recombination Hotspot Details

Recombination hotspots are short regions of the genome with high recombination rate relative to the background. As the recombination rate between two DNA locations tunes the correlation between their corresponding genealogies, hotspots play an important role in complex disease inheritance patterns. In order to develop accurate methodology, a precise mathematical definition of a hotspot needs to be specified in accordance with the signatures of biological interest. We use the following:

**Definition 1** (Recombination Hotspot)**.** Let a window over the genome be subdivided into three subwindows $w = (w_l, w_h, w_r)$ with physical distances (i.e., window widths) $\alpha_l, \alpha_h$, and $\alpha_r$, respectively, where $w_l, w_h, w_r \in \mathcal{G}$ where $\mathcal{G}$ is the space over all possible subwindows of the genome. Let a mean recombination map $R : \mathcal{G} \to \mathbb{R}_+$ be a function that maps from a subwindow of the genome to the mean recombination rate per base pair in the subwindow. A recombination hotspot for a given mean recombination map $R$ is a window $w$ which satisfies the following properties:

1. Elevated local recombination rate: $R(w_h) > k \cdot \max\big(R(w_l), R(w_r)\big)$

2. Large absolute recombination rate: $R(w_h) > k\tilde{r}$

where $\tilde{r}$ is the median (at a per base pair level) genome-wide recombination rate, and $k > 1$ is the relative hotspot intensity.

The first property is necessary to enforce the locality of hotspots and rule out large regions of high recombination rate, which are typically not considered hotspots by biologists. The second property rules out regions of minuscule background recombination rate in which sharp relative spikes in

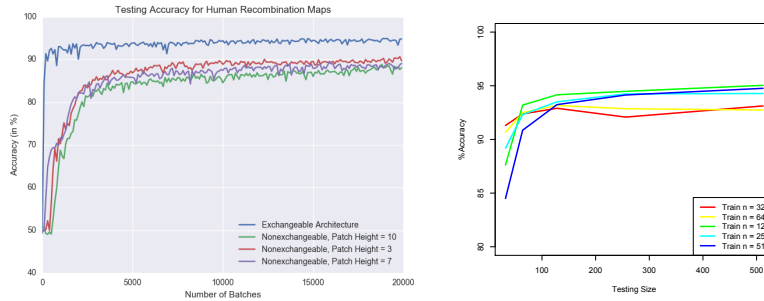

Figure 2: (Left)Accuracy comparison between exchangeable vs nonexchangeable architectures. (Right)Performance of changing the number of individuals at test time for varying training sample sizes.

recombination still remain too small to be biologically interesting. The median is chosen here to be robust to the right skew of the distribution of recombination rates. Typically, for the human genome we use $\alpha_l = \alpha_r = 13$ kb, $\alpha_h = 2$ kb, and $k = 10$ based on experimental findings.

## 5.2 Evaluation of Exchangeable Neural Network

We compare the behavior of an explicitly exchangeable architecture to a nonexchangeable architecture that takes 2D convolutions with varying patch heights. The accuracy under human-like population genetic parameters with varying 2D patch heights is shown in the left panel of Figure 2. Since each training point is simulated on-the-fly, data augmentation is performed implicitly in the nonexchangeable version without having to explicitly permute the rows of each training point. As expected, directly encoding the permutation invariance leads to more efficient training and higher accuracy while also benefiting from a faster per-batch computation time. Furthermore, the slight accuracy decrease when increasing the patch height confirms the difficulty of learning permutation invariance as $n$ grows. Another advantage of exchangeable architectures is the robustness to the number of individuals at test time. As shown in right panel of Figure 2, the accuracy remains above 90% during test time for sample sizes roughly 0.1–20× the train sample size.

## 5.3 Evaluation of Simulation-on-the-fly

Next, we analyze the effect of simulation-on-the-fly in comparison to the standard fixed training set. A fixed training set size of 10000 was used and run for 20000 training batches and a test set of size 5000. For a network using simulation-on-the-fly, 20000 training batches were run and evaluated on the same test set. In other words, we ran both the simulation on-the-fly and fixed training set for the same number of iterations with a batch size of 50, but the simulation-on-the-fly draws a fresh datapoint from the generative model upon each update so that no datapoint is used more than once. The weights were initialized with a fixed random seed in both settings with 20 replicates. Figure 3 (left) shows that the fixed training set setting has both a higher bias and higher variance than simulation-on-the-fly. The bias can be attributed to the estimation error of a fixed training set in which the empirical risk surface is not a good approximation of the population risk surface. The variance can be attributed to an increase in the number of poor quality local optima in the fixed training set case.

We next investigated posterior calibration. This gives us a measure for whether there is any bias in the uncertainty estimates output by the neural network. We evaluated the calibration of simulation-on-the-fly against using a fixed training set of 10000 datapoints. The calibration curves were generated by evaluating 25000 datapoints at test time and binning their posteriors, computing the fraction of true labels for each bin. A perfectly calibrated curve is the dashed black line shown in Figure 3 (right). In accordance with the theory in Section 3.3, the simulation-on-the-fly is much better calibrated with an increasing number of training examples leading to a more well calibrated function. On the other hand, the fixed training procedure is poorly calibrated.

## 5.4 Comparison to LDhot

We compared our method against LDhot in two settings: (i) sampling empirical recombination rates from the HapMap recombination map for CEU and YRI (i.e., Yoruba in Ibadan, Nigeria) [31] to set the

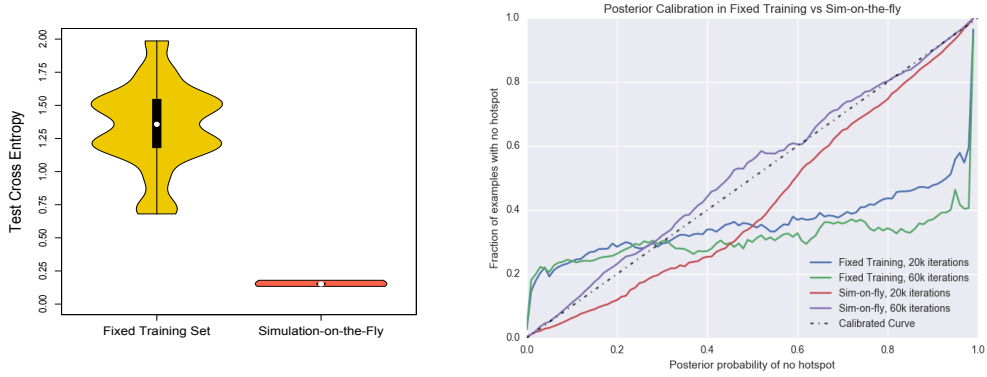

Figure 3: (Left)Comparison between the test cross entropy of a fixed training set of size 10000 and simulation-on-the-fly. (Right)Posterior calibration. The black dashed line is a perfectly calibrated curve. The red and purple lines are for simulation-on-the-fly after 20k and 60k iterations; the blue and green lines for a fixed training set of 10k points, for 20k and 60k iterations.

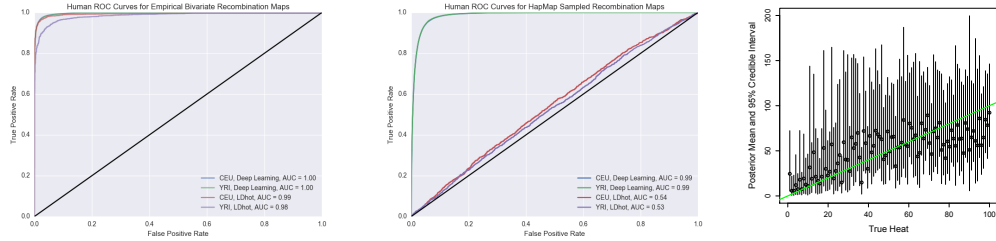

Figure 4: (Left) ROC curve in the CEU and YRI setting for the deep learning and `LDhot` method. The black line represents a random classifier. (Middle) Windows of the HapMap recombination map drawn based on whether they matched up with our hotspot definition. The blue and green line coincide almost exactly. (Right) The inferred posteriors for the continuous case. The circles represent the mean of the posterior and the bars represent the $95\%$ credible interval. The green line shows when the true heat is equal to the inferred heat.

background recombination rate, and then using this background to simulate a flat recombination map with $10 - 100\times$ relative hotspot intensity, and (ii) sampling segments of the HapMap recombination map for CEU and YRI and classifying them as hotspot according to our definition, then simulating from the drawn variable map.

The ROC curves for both settings are shown in Figure 4. Under the bivariate empirical background prior regime where there is a flat background rate and flat hotspot, both methods performed quite well as shown on the left panel of Figure 4. We note that the slight performance decrease for YRI when using `LDhot` is likely due to hyperparameters that require tuning for each population size. This bivariate setting is the precise likelihood ratio test for which `LDhot` tests. However, as flat background rates and hotspots are not realistic, we sample windows from the HapMap recombination map and label them according to a more suitable hotspot definition that ensures locality and rules out neglectable recombination spikes. The middle panel of Figure 4 uses the same hotspot definition in the training and test regimes, and is strongly favorable towards the deep learning method. Under a sensible definition of recombination hotspots and realistic recombination maps, our method still performs well while `LDhot` performs almost randomly. We believe that the true performance of `LDhot` is somewhere between the first and second settings, with performance dominated by the deep learning method. Importantly, this improvement is achieved without access to any problem-specific summary statistics.

Our approach reached $90\%$ accuracy in fewer than 2000 iterations, taking approximately $0.5$ hours on a $64$ core machine with the computational bottleneck due to the `msprime` simulation [29]. For `LDhot`, the two-locus lookup table for variable population size using the `LDpop` fast approximation

[32] took 9.5 hours on a 64 core machine (downsampling $n = 198$ from $N = 256$). The lookup table has a computational complexity of $O(n^3)$ while per-iteration training of the neural network scales as $O(n)$, allowing for much larger sample sizes. In addition, our method scales well to large local regions, being able to easily handle 800-SNP windows.

### 5.5 Recombination Hotspot Intensity Estimation: The Continuous Case

To demonstrate the flexibility of our method in the continuous parameter regime, we adapted our method to the problem of estimating the intensity (or heat) of a hotspot. The problem setup fixes the background recombination rate $R(w_l) = R(w_r) = 0.0005$ and seeks to estimate the relative hotspot recombination intensity $k$. The demography is set to that of CEU. The hotspot intensity $k$ was simulated with a uniform distributed prior from 1 to 100.

For continuous parameters, arbitrary posteriors cannot be simply parameterized by a vector with dimension in the number of classes as was done in the discrete parameter setting. Instead, an approximate posterior distribution from a nice distribution family is used to get uncertainty estimates of our parameter of interest. This is achieved by leveraging our exchangeable network to output parameter estimates for the posterior distribution as done in [33]. For example, if we use a normal distribution as our approximate posterior, the network outputs estimates of the mean and precision. The corresponding loss function is the negative log-likelihood

$$-\log p(k|\mathbf{x}) = -\frac{\log \tau(\mathbf{x})}{2} + \frac{\tau(\mathbf{x})(k - \mu(\mathbf{x}))^2}{2} + \text{const}, \tag{3}$$

where $\mu$ and $\tau$ are the mean and the precision of the posterior, respectively. More flexible distribution families such as a Gaussian mixture model can be used for a better approximation to the true posterior.

We evaluate our method in terms of calibration and quality of the point estimates to check that our method yields valid uncertainty estimates. The right panel of Figure 4 shows the means and 95% credible intervals inferred by our method using log-normal as the approximate posterior distribution. As a measure of the calibration of the posteriors, the true intensity fell inside the 95% credible interval 97% of the time over a grid of 500 equally spaced points between $k = 1$ to 100. We measure the quality of the point estimates with the Spearman correlation between the 500 equally spaced points true heats and the estimated mean of the posteriors which yielded 0.697. This was improved by using a Gaussian mixture model with 10 components to 0.782. This illustrates that our method can be easily adapted to estimate the posterior distribution in the continuous regime.

## 6 Discussion

We have proposed the first likelihood-free inference method for exchangeable population genetic data that does not rely on handcrafted summary statistics. To achieve this, we designed a family of neural networks that learn an exchangeable representation of population genetic data, which is in turn mapped to the posterior distribution over the parameter of interest. Our simulation-on-the-fly training paradigm produced calibrated posterior estimates. State-of-the-art accuracy was demonstrated on the challenging problem of recombination hotspot testing.

The development and application of exchangeable neural networks to fully harness raw sequence data addresses an important challenge in applying machine learning to population genomics. The standard practice to reduce data to ad hoc summary statistics, which are then later plugged into a standard machine learning pipelines, is well recognized as a major shortcoming. Within the population genetic community, our method proves to be a major advance in likelihood-free inference in situations where ABC is too inaccurate. Several works have applied ABC to different contexts, and each one requires devising a new set of summary statistics. Our method can be extended in a black-box manner to these situations, which include inference on point clouds and quantifying evolutionary events.

### Acknowledgements

We thank Ben Graham for helpful discussions and Yuval Simons for his suggestion to use the decile. This research is supported in part by an NSF Graduate Research Fellowship (JC); EPSRC grants EP/L016710/1 (VP) and EP/L018497/1 (PJ); an NIH grant R01-GM094402 (JC, JPS, SM, and YSS); and a Packard Fellowship for Science and Engineering (YSS). YSS is a Chan Zuckerberg Biohub

investigator. We gratefully acknowledge the support of NVIDIA Corporation with the donation of the Titan X Pascal GPU used for this research. This research also used resources of the National Energy Research Scientific Computing Center, a DOE Office of Science User Facility supported by the Office of Science of the U.S. Department of Energy under Contract No. DE-AC02-05CH11231.

## Footnotes

[1]Sites that have $> 2$ bases are rare and typically removed. Thus, a binary encoding can be used.

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
