[Supplementary Material]

# Supplementary material for the paper:
# A Likelihood-Free Inference Framework for Population Genetic Data using Exchangeable Neural Networks

**Jeffrey Chan**
University of California, Berkeley
chanjed@berkeley.edu

**Valerio Perrone**
University of Warwick
v.perrone@warwick.ac.uk

**Jeffrey P. Spence**
University of California, Berkeley
spence.jeffrey@berkeley.edu

**Paul A. Jenkins**
University of Warwick
p.jenkins@warwick.ac.uk

**Sara Mathieson**
Swarthmore College
smathie1@swarthmore.edu

**Yun S. Song**
University of California, Berkeley
yss@berkeley.edu

## A  Simulation Details for Recombination Hotspot Testing

### A.1  Set-up

We encode population genetic data $\mathbf{x}$ as follows. Let $\mathbf{x}_S$ be the binary $n \times d$ matrix with 0 and 1 as the common and rare nucleotide variant, respectively, where $n$ is the number of sequences, and $d$ is the number of SNPs. Let $\mathbf{x}_D$ be the $n \times d$ matrix storing the distances between neighboring SNPs, so each row of $\mathbf{x}_D$ is identical and the rightmost distance is set to 0. Define $\mathbf{x}$ as the $n \times d \times 2$ tensor obtained by stacking $\mathbf{x}_S$ and $\mathbf{x}_D$. To improve the conditioning of the optimization problem, the distances are normalized such that they are on the order of $[0, 1]$.

The standard generative model for such data is the coalescent, a stochastic process describing the distribution over genealogies relating samples from a population of individuals. The coalescent with recombination [1, 2] extends this model to describe the joint distribution of genealogies along the chromosome. The recombination rate between two DNA locations tunes the correlation between their corresponding genealogies. Population genetic data derived from the coalescent obeys translation invariance along a sequence conditioned on local recombination and mutation rates which are also translation invariant. In order to take full advantage of parameter sharing, our chosen architecture is given by a convolutional neural network with tied weights for each row preceding the exchangeable layer, which is in turn followed by a fully connected neural network.

### A.2  Recombination Hotspot Testing

Recombination hotspots are short regions of the genome ($\approx 2$ kb in humans) with high recombination rate relative to the background recombination rate. To apply our framework to the hotspot detection problem, we define the overall graphical model in Figure 1. The shaded nodes represent the observed variables. Denote $w$ as a small window (typically $< 25$ kb) of the genome such that $X_w$ is the population genetic data in that window, and $X_{-w}$ is the rest. Similarly, let $\rho_w$ and $\rho_{-w}$ be the recombination map in the window and outside of the window, respectively. While $\rho_w$ and $\rho_{-w}$ have a weak dependence (dashed line) on $X_{-w}$ and $X_w$ respectively, this dependence decreases rapidly and is

ignored for simplicity. More precisely, weak dependence means that $P(\rho_w, X_{-w}) \approx P(\rho_w)P(X_{-w})$ as shown in Equation 3.1 of [3] via a Taylor expansion argument. The intuition for this is that $\rho$ tunes the correlation between neighboring sites so each site is effectively independent of recombination rates at distal sites.

Let $q$ be the relative proportion of the sample possessing each mutation, and $\eta$ be the population size function. Intuitively, $\eta$ determines the rate at which the genealogies (can be thought of as binary trees) branch. $q$ is a summary statistic of $\eta$ which we observe that allows us to fix the population size in an empirical Bayes style throughout training for simplicity using `SMC++`.

Let $\theta$ be the mutation rate and $h$ be the indicator function for whether the window defines a hotspot. Conditioned on $q$, $\eta$ is only weakly dependent on $X_w$.

We define our prior as follows. We sample the hotspot indicator variable $h \sim$ Bernoulli$(0.5)$ and the local recombination maps $\rho_w \sim \hat{P}(\rho_w \mid h)$ from the released fine-scale recombination maps of HapMap [4]. The human mutation rate is fixed to that experimentally found in [5]. Since `SMC++` is robust to changes in any small fixed window, inferring $\hat{\eta}$ from $X$ has minimal dependence on $\rho_w$.

To test for recombination hotspots:

1. Simulate a batch of $h$ and $\rho_w$ from the prior and $X_w$ from `msprime` [6] given $h$ and $\rho_w$.
2. Feed a batch of training examples into the network to learn $\mathbb{P}(h \mid X_w)$.
3. Repeat until convergence or for a fixed number of iterations.
4. At test time, slide along the genome to infer posteriors over $h$.

Figure 1: Graphical model of recombination hotspot inference: $\theta$ is the mutation rate, $\eta$ the population size function, $q$ the relative proportion of the sample possessing each mutation, $\rho_{-w}$ the recombination rate function outside of the window, $\rho_w$ the recombination rate function inside the window, $h$ whether the window is a hotspot, $X_{-w}$ the population genetic data outside of the window, and $X_w$ the data inside the window. The dashed line signifies that, conditioned on $q$, $\eta$ is weakly dependent on $X_w$ for suitably small $w$, and $\rho_{-w}$ and $\rho_w$ are only weakly dependent on $X_w$ and $X_{-w}$.

## B  Statistical Properties of Our Method: Proofs

**Proof of Proposition 1**  By the Universal Approximation Theorem and the interpretation of simulation-on-the-fly as minimizing the expected KL divergence between the population risk and the neural network, the training procedure minimizes the objective function for any $\mathbf{x}$, $\epsilon > 0$, $\delta > 0$, we can pick a $H > H_0$, and $N > N_0$ such that,

$$\min_{\mathbf{w}} \mathbb{E}_{\mathbf{x}} \left[ KL \Big( \mathbb{P}(\theta \mid \mathbf{x}) \,\|\, \mathbb{P}_{DL}^{(N)}(\theta \mid \mathbf{x}; \mathbf{w}, H) \Big) \right] < \epsilon.$$

Let $\mathbf{w}^*$ be a minimizer of the above expectation. By Markov's inequality, we get for every $\mathbf{x}$ and $\delta > 0$ such that for all $H > H_0$ and $N > N_0$

$$KL \Big( \mathbb{P}(\theta \mid \mathbf{x}) \,\|\, \mathbb{P}_{DL}^{(N)}(\theta \mid \mathbf{x}; \mathbf{w}^*, H) \Big) < \delta$$

with probability at least $1 - \frac{\epsilon}{\delta}$. □

**Proof of Corollary 1**  As above, for any $\mathbf{x}$, $\epsilon > 0$, $\delta > 0$, there exists a $H > H_0$, and $N > N_0$ such that

$$\min_{\mathbf{w}} \mathbb{E}_{\mathbf{x}} \left[ KL \Big( \mathbb{P}(\theta \mid \mathbf{x}) \parallel \mathbb{P}_{DL}^{(N)}(\theta \mid \mathbf{x}; \mathbf{w}, H) \Big) \right] < \epsilon.$$

Furthermore, for all $\mathbf{x}$, the KL is bounded at the minimizer since $\mathbb{P}(\mathbf{x}) > 0$ for all $\mathbf{x}$ resulting in the following bound

$$KL \Big( \mathbb{P}(\theta \mid \mathbf{x}) \parallel \mathbb{P}_{DL}^{(N)}(\theta \mid \mathbf{x}; \mathbf{w}^*, H) \Big) < \max_{\mathbf{x}} \frac{\epsilon}{\mathbb{P}(\mathbf{x})}$$

independent of $\mathbf{x}$. Thus, the training procedure results in a function mapping that uniformly converges to the posterior $\mathbb{P}(\theta \mid \mathbf{x})$. □

## C    LDhot details

The most widely-used technique for recombination hotspot testing is `LDhot` as described in [7]. The method performs a generalized composite likelihood ratio test using the two-locus composite likelihood based on [8] and [9]. The composite two-locus likelihood approximates the joint likelihood of a window of SNPs $w$ by a product of pairwise likelihoods

$$CL(\rho \mid \mathbf{x}) = \prod_{1 \leq |i-j| \leq z} L(\rho_{ij} \mid \mathbf{x}_{ij}),$$

where $X_{ij}$ denotes the data restricted only to SNPs $i$ and $j$, and $\rho_{ij}$ denotes the recombination rate between those sites. Only SNPs within some distance, say $z = 50$, are considered.

Two-locus likelihoods are computed via an importance sampling scheme under a constant population size ($\eta = 1$) as in [9]. The likelihood ratio test uses a null model of a constant recombination rate and an alternative model of a differing recombination rate in the center of the window under consideration:

$$\Lambda = -2 \log \left( \frac{\sup_{\rho_{\text{hot}}, \rho_{\text{bg}}} CL(\rho_{\text{hot}}, \rho_{\text{bg}} \mid X)}{\sup_{\rho_{\text{const}}} CL(\rho_{\text{const}} \mid X)} \right).$$

The two-locus likelihood can only be applied to a single population with constant population size, constant mutation rate, and without natural selection. Furthermore, the two-locus likelihood is an uncalibrated approximation of the true joint likelihood. In addition, [10] and [7] performed simulation studies showing that LDhot has good power but their simulation scenarios were unrealistic because its null hypothesis leads to a comparison against a biologically unrealistic flat background rate. In order to fairly compare our likelihood-free approach against the composite likelihood-based method in realistic human settings, we extended the `LDhot` methodology to apply to a piecewise constant population sizes using two-locus likelihoods computed by the software `LDpop` [11]. Unlike the method described in [10], our implementation of `LDhot` uses windows defined in terms of SNPs rather than physical distance in order to measure accuracy via ROC curves, since the likelihood ratio test is a function of number of SNPs. Note that computing the approximate two-locus likelihoods for a grid of recombination values is at least $O(n^3)$, which could be prohibitive for large sample sizes.

## D    Additional Experiments

**Regularization**    The simulation-on-the-fly paradigm obviates the need for modern regularization techniques such as dropout. This is due to the fact that there is no notion of overfitting since each training point is used only once and a large number of examples are drawn from the population distribution. As shown in Figure 2(left), dropout does not help improve the accuracy of our method and, in fact, leads to a minor decrease in performance. As expected, directly optimizing the population risk minimizer circumvents the problem of overfitting.

**Phasing**    Often times in sequencing data it is difficult to separate the DNA contributions from each chromosome (we have two — one from each parent). Thus, data is typically expressed as a sum so that $\mathbf{x} \in \{0, 1, 2\}^d$. Most population genetic methods require the data to be separated, referred to as *phased*. Phasing algorithms can often introduce significant bias into downstream inference, so methods that do not require phased data are particularly useful. Our approach can

Figure 2: (Left)A comparison of different dropout rates. Dropout has a minimal (or slightly negative) effect on test accuracy under the simulation-on-the-fly regime. (Right)Accuracy comparison between unphased and phased data.

flexibly perform inference directly on phased or unphased data, the latter being a challenge for model-based approaches. Inference directly on unphased data allows us to implicitly integrate over possible phasings, reducing the bias introduced by fixing the data to a single phasing. In the case of recombination hotspots, we have found only a minor decrease in accuracy for small sample sizes corresponding to the reduction in statistical signal when inference is performed on unphased data. We quantified the effect of having accurately phased (haploid) data in comparison to unphased(diploid) data. Specifically, inference was run by simulating haploid data and randomly pairing them to construct diploid data such that the height of the diploid matrix is half that of the haploid matrix. We ran the experiment for $n = 16, 32, 64$ as shown in Figure 2(right) and found that the our method is robust, remaining highly accurate for unphased data.