[Reviews · NeurIPS 2018]

Reviewer 1



In population genetics, a common problem is the inference of evolutionary events and parameters given some observed DNA sequencing data. This is typically done by leveraging genetic simulators based on the coalescence model. For example, in approximate Bayesian Computation (ABC) one can infer the posterior distribution of the evolutionary parameter of interest by (i) drawing its value from a prior, (ii) generate population genetic data from a simulator using the drawn parameter value and (iii) weight the importance of the sample based on the similarity between the simulated and the real data. Similarity is typically defined in terms of a pre-chosen set of summary statistics. The authors propose an inference strategy that does not rely on any choice of summary statistics. Briefly, the authors propose to use a neural network to invert the generative process underlying the coalescent simulator. Specifically, one can use steps (i) and (ii) as in ABC to generate multiple pairs (parameter value, population genetic data), and then train a neural network that can predict the parameter value given the genetic data. After training, a posterior can be computed by forward passing the real data through the network. As training data is only based on simulators, the authors propose a simulation-on-the-fly method where each training point is generated afresh. Finally, the authors consider an exchangeable neural network architecture to account for the exchangeability of individual genetic sequences in the population (in the case of unrelated individuals). In an application to the recombination hotspot testing problem, the authors demonstrate the advantages of employing a “simulation-on-the-fly” strategy and an exchangeable architecture. Finally, they show that their approach competes with a state-of-the-art method for recombination hotspot testing. The method introduced in this paper is novel and interesting and can in principle be extended to tackle different problems in population genetics. However, I have some points I would like to bring up: 1) The authors say that the neural network is used to learn a posterior of the evolutionary parameters. While this is not wrong, I think it may be confusing. If my understanding is correct, I think the neural network is mostly used to invert the generative process underlying the coalescent simulator. It is trained using simulated data (where the groundtruth value of the parameter to infer is known) while after training, it can be used to “invert the real data into a posterior on the parameter of interest”. 2) The authors should make more explicit which is the loss of the exchangeable neural network. My understanding is that in most setting the evolutionary parameters can take a finite number of discrete values and that in this case, a cross-entropy loss would be used. Is this correct? Also, what do the authors recommend if the parameter to infer is continuous? 3) In the recombination hotspot testing problem, the neural network is trained to classify 20-SNP windows in positive/negative cases of recombination hotspots. While the window size may be fine for this inference problem, this choice could be suboptimal when inferring other parameters. Of course, one could increase the window size, however, the number of weights in the dense layers would also increase (if the same neural network architecture is the same). Do the authors have any comment on this? Minor: - Related to point 1, in Section 3.1, I think that the authors should remind to the reader that there are observed genetic data and that the goal is to infer the posterior of some evolutionary parameter given the observed data. - Which test statistic is used to obtain the ROC curves in in figure 4 for the two methods? I raised my score from 7 to 8 after the reading the response from the authors.

Reviewer 2



In this paper the authors introduce a framework for inference in population genetic models with a focus on recombination hotspot identification, composed of two main ingredients: (1) “simulation-on-the-fly” whereby datasets with known parameters are simulated (under the coalescent) with the model trained on each dataset exactly once, and (2) an exchangeable neural network that maps from the simulated data to the posterior distribution over parameter values. The observed data can then be used as input in a similar manner to recognition networks to infer the posterior distribution for the parameters given the observed data. The authors demonstrate that theoretically the posterior is well-calibrated and show that their method outperforms state-of-the-art for recombination hotspot discovery. The proposed method appears to perform excellently on detecting recombination hotspots and could easily be considered state-of-the-art if a more comprehensive evaluation was undertaken. Of particular note is the ability of the model to perform amortized inference on unseen datasets, potentially meaning it could be distributed as a pre-compiled model. That said - I think the overall claim that the authors have developed a general framework for likelihood-free inference from population genetic data is untrue. In particular, it is not obvious how the method extends to continuous parameters, and is only demonstrated in the context of recombination hotspot testing. Further comments are below. The notation in the problem setup (section 3.1) is unclear. Does the index (i) introduced on line 101 reference each simulated dataset? If so this should be stated. Further, X is a dataset-by-sample-by-SNP tensor (array) while bold-x is a sample-by-snp matrix, while unbold-x is a vector of SNPs. This notation is quite unstandard (ie bold-x would be a vector, etc), and while this is a very minor point, it makes for confused reading. On line 114 the authors introduce the exchangeable neural network that learns the posterior distribution over parameters given a simulated dataset. In the example they provide, this is essentially a classification problem of whether a section is a hotspot or not, and thus the representation of the posterior is as a softmax over two values per window. However, this limits the generality of the proposed method - how does it work if the quantity for which I wish to derive a posterior distribution is continuous? Then such a neural net would input a simulated dataset and output a point estimate rather than a distribution. This problem could be circumvented for categorical variables (though with a large increase in the overall number of parameters required if one-hot encoded), but it is not obvious how this extends to continuous variables in this setting (e.g. mutation rates), which conflicts with the authors’ claim in the abstract that implies the method is “general-purpose”. In section 5.3 the authors compare their simulation on the fly approach (new dataset per epoch) to a fixed training set size. In the fixed approach they state that a training set size of 10000 was used while in the on-the-fly approach 20000 batches were used, which I assume means 20000 data sets? If this is the case then the on-the-fly approach has used twice as much data, which could easily explain the increased test-set performance? I may be misunderstanding the setup here. Line 233 - \alpha are defined as “physical distances” - are these the widths of the windows? Line 238 - k probably needs defined as k > 1

Reviewer 3



The authors consider using neural networks (NN) to learn a mapping from data to parameters for population genetics models. Such models typically have highly intractable likelihoods as a result of their complex latent structure. However, sampling from these models is relatively straightforward. As a result a typical approach has been to use Approximate Bayesian Computation which compares user-specified statistics from the simulation and real data to decide if the sampled parameters are "good". The choice of statistic is clearly critical however and relies on expert knowledge for a specific model. There has been work using NNs to output parameters based on a set of summary statistics. The current paper proposes using NNs where the input is the raw genotype data itself. Since this data is exchangeable (the labeling of the individuals doesn't matter) it is logical to consider exchangeable NNs (ENN) that leverage this structure to learn more efficiently. The authors construct the following ENN: a) apply the same (convolutional) NN to each individual's genotype vector b) apply a symmetric function (specifically element-wise top decile) to collapse across individuals c) apply a final dense NN. The idea of performing "simulation-on-the-fly" is discussed, i.e. generating every new training sample from the generative model so that estimation error is small. Results are presented on recombination hotspot testing, with improvements over the state-of-the-art at comparable or lower computational cost. The paper is well written and clear. I work in genetics so recombination hotspots are a familiar concept: I wonder if they are sufficiently motivated for a general ML audience? Also the term "calibrated posterior" is used extensively but not defined. I would guess this means an estimate of the posterior whose variance is an unbiased estimate of the variance of the true posterior? (noting that is dependent on the choice of generative model). The motivation, theory and experiments successfully demonstrate the value of the approach relative to current approaches in the field. The key question to me is how important is the choice of symmetric function? A wide class of symmetric functions could be used, and indeed the function itself could be learnt - although the suggestions are all fixed functions. It would be straightforward to test a few possibilities empirically (maybe the authors did this before picking top decile?): presumably in practice one would want to include this as part of hyperparameter/architechure tuning. In typical uses of such an approach we will have a dataset we are particularly interested in. In that case we don't care so much about the general accurarcy of the method but instead its accuracy at this one dataset. This suggests to me that there should be potential gains by focusing training on regions of parameter space that produce data "close" to the real data. I suspect this would look something like a supervised VAE. I'd be interested to hear the authors thoughts. Finally I'm curious about how large a region of genome the authors envisage modeling simultaneously. Recombination hotspot testing is a case where it makes sense to look at small genomic regions. However, if you were trying to look at something like demographic history there might be signal coming from the entire genome. Would you divide up into small regions and somehow aggregate predictions? Overall I think this is a valuable contribution to the pop gen field and is hopefully of interest to a wider ML audience. There are some questions I would like to see addressed but I realize it is difficult to cover all yours bases in a conference paper. After the reading the author response I'm uping my score to top 50% of papers.